# Morphological Characteristics of Dimples of Ductile Fracture of VT23M Titanium Alloy and Identification of Dimples on Fractograms of Different Scale

**DOI:** 10.3390/ma12132051

**Published:** 2019-06-26

**Authors:** Ihor Konovalenko, Pavlo Maruschak, Janette Brezinová, Jozef Brezina

**Affiliations:** 1Department of Industrial Automation, Ternopil National Ivan Puluj Technical University, Rus’ka str. 56, 46001 Ternopil, Ukraine; 2Department of Engineering Technologies and Materials, Faculty of Mechanical Engineering, Technical University of Košice, Mäsiarska 74, 040 01 Košice, Slovakia; 31st Surgical Clinic of Faculty of Medicine Pavol Jozef Šafárik University in Košice and University Hospital of L. Pasteur in Košice, 04022 Košice, Slovakia

**Keywords:** failure analysis, fracture mechanisms, titanium alloy, shape and size of dimples

## Abstract

The authors developed a method for the automated detection and calculation of quantitative parameters of dimples of ductile fracture on the digital images of fracture surfaces obtained at different scales. The processing algorithm of fractographic images was proposed, which allowed high quality recognition of the shape and size of dimples to be achieved, taking into account the morphological features of their digital images. The developed method for identifying dimples of various physical and morphological characteristics was tested on the VT23M alloy. The test results showed that the method meets the quality requirements for the automated diagnostics of fracture mechanisms of titanium alloys.

## 1. Introduction

Titanium alloys are used for the protective clothing of soldiers (plates of body armor), as well as implants, surgical instruments, internal and external prosthetics, including such critical high-tech elements as heart valves. In addition, titanium alloys are used for the manufacture of aircrafts and space vehicles [1,2,3,4,5].

Fracture patterns of metals in general and high-strength titanium alloys in particular depend on a stage-like nature of damage accumulation during deformation. That is, macro-fracture patterns actually depend on the stage-like nature of transition from the equilibrium to non-equilibrium accumulation of damage, and on the random stage of fracture [6,7]. Deformation causes a plurality of pores concentrated predominantly in the central part of the neck, which coalesce with grain conglomerates leading to the formation of a continuous crack in the plane normal to the direction of loading. Thus, the central crack, which grows by means of thinning and breaking connections between the pores, together with the newly formed crack leave “traces” on the surface in the form of dimples, which are a sort of “markers” or signs that can be used to reproduce the “history” of the material fracture [8,9,10].

It is known that the processes of structural damage are depicted on the stress–strain diagram causing its nonlinearity, whereas fracture can be considered as the final stage of deformation that goes from the micro- and meso- levels to the macrolevel [11]. These physical–mechanical links between the levels of deformation and fracture are particularly important when analyzing the causes of unpredictable fracture of structures and evaluating their degradation mechanisms using fractographic analysis.

Fractographic analysis, as a separate line of research in fracture mechanics, uses physics of solid body (the theory of dislocations, etc.), material science (the construction of correlations between fracture parameters and the size of grains, inclusions, etc.), optic-digital methods, etc., as its tools [12,13,14,15]. Until recently, parameter measurements of fracture surfaces were made manually or automatically, however, measurement software was positioned by the operator, which complicated fractographic research. A large variety of materials complicated not only the development of generalized approaches to the morphological analysis of destroyed surfaces, but also the interpretation of the data obtained [16,17].

It should be noted that in the previous works, the authors proposed approaches that formed the basis for the development of the method of fractographic recognition, control and calculation of parameters of the dimples of tearing based on the use of neural networks [18,19,20]. In previous work, 17 models of neural networks with various sets of hyperparameters have been developed and investigated. Then, their accuracy and speed were investigated, and the optimal neural network model was selected. The proposed network contained two convolution layers, two subsampling layers, and two fully connected layers (Figure 1).

In the developed network, the linear activation function Rectified Linear Unit (ReLU) was used. As a loss function, categorical cross-entropy was used. It measures the performance of a classification model, of which the output is a probability value between 0 and 1. As metrics, we used accuracy: the proportion of correct predictions with respect to the targets. During training of CNNs, we used the Adam optimizer. It implements an algorithm for first-order gradient-based optimization of stochastic objective functions, based on adaptive estimates of lower-order moments.

For training of the neural network, we used 562,856 image fragments measuring 31 × 31 pixels. The test dataset consisted of 134,596 image fragments.

The neural network classified the pixels into two categories: “dimples” and “edges”, and admitted the affiliation of each pixel to one of these classes. Also, we compared the results of dimples recognition by the proposed network with results obtained by a known algorithm to analyze the fractograms. It has been found that the results are very similar (more than 90% similarity), but the neural network reveals the necessary features more accurately than the previous method [19,20,21].

However, the question arises as to how the area of fractographic control affects the calculated parameters of dimples. It remains relevant to create effective computer diagnostic systems for the automated processing of the fractographic images obtained, preceded by a diagnostic conclusion on the surface condition. Therefore, the need to develop and test the approaches of automated fractographic analysis of material fractures and their interpretation, in our opinion, does not raise doubts.

The purpose of this research is to develop and investigate the method for recognizing fractographic images, and find scientific approaches to the analysis of coalescence processes undergone by dimples of ductile tearing, taking into account their size and geometry.

## 2. Methods of Macro-Analysis of Fracture Surfaces

High-strength titanium alloy VT23M was used as a research material. Specimens were loaded with a dynamic impulse and then stretched to fracture statically. The process of mechanical testing is described in detail in [6,7]. Investigation of the fracture surface was performed by scanning electron microscopy using the scanning electron microscope (SEM) REM 106I.

Within the framework of the morphological analysis of fractures of specimens from the VT23M alloy, the fracture surface was considered as a set of complementary elements, the main elements of which are the dimples of tearing and boundaries between them [22,23]. It should be noted that such interpretation not only provides for the physical and mechanical correctness of further analysis of the size and shape of dimples of tearing, but allows the process of deformation and fracture of the VT23M alloy to be considered as a multi-stage process taking place at different scale levels.

In the macroscopic view of the fracture surface of a flat specimen from the VT23M alloy, two zones are observed: the fibrous zone (the central part of the specimen) and the zone of the cut (in the vicinity of the lateral surfaces) indicating fracture of the VT23M alloy in the conditions of stable crack propagation. Characteristic microphotographs demonstrate the presence of a dimple relief on the fracture surface of the VT23M alloy at different magnifications, as shown in Figure 2a.

The images presented in Figure 2a show a fragment of the fracture surface and the geometry of dimples. The images presented in Figure 2b–d show the same surface (outlined by a frame) located at the cut of the specimen at different magnifications (outlined by the frame). It should be noted that in all cases these are dimples of tearing with fairly clear-cut edges and a “flat” bottom.

Various magnifications of the fracture surface allow characteristic fracture mechanisms to be established at various scale levels and peculiarities of the development of defects, in particular, macro-fracture of specimens at the supercritical stage, which immediately precedes the moment of fracture [24,25].

The investigated fracture surface has a dimple structure formed as a result of coalescence of a set of micropores. The localization and identification of dimples in the image show that the appearance of pores was accompanied by plastic deformation of the intersections between them [18,19]. However, they are thick enough to identify each individual dimple. The crests of tearing between the pores have a matted uneven surface, which differs by the color intensity at different parts of the fracture. The fracture surface is uneven; smaller dimples are located on the lateral surfaces of large dimples, which are marked by crests of tearing [26,27].

On the basis of the previously proposed algorithm for identifying and investigating dimples, the parameters of dimples [28] found on the fracture surface were evaluated, and for each of them, the equivalent diameter (*d_i_*), maximal length, visual depth, inclination and area were calculated. The equivalent diameter of the dimple is the diameter of the circle with an area equal to the recognized area of the dimple. 

The nature of the surface damage accumulation was determined by the parameters of the dimples of tearing located on it [8,29]. It is established empirically that the dimple shape coefficient is informative for their classification. In order to detect changes in the shape of dimples, the roundness coefficient *K_c_* was used, which is equal to the percentage of dimple pixels that fall into a circle with an equivalent diameter di, the center of which is aligned with the center of mass of dimple *C_i_(x_ci_, y_ci_)* [18,19,20]:(1)$$K_{c}=\frac{\sum_{m=1}^{f_{i}}g\left(\vec{r_{m}},d_{i}\right)}{f_{i}}\cdot 100\%$$
where $g\left(\vec{r_{m}},D_{i}\right)$ is the indicative function that shows whether the m-th pixel falls into the circle with equivalent diameter *d_i_*;
(2)$$g\left(\vec{r_{m}},d_{i}\right)=\{\begin{array}{c} 1,\;\mathrm{at}\;\left|\vec{r_{m}}\right|\leq d_{i}/2\\ 0,\;\mathrm{at}\;\left|\vec{r_{m}}\right|>d_{i}/2 \end{array}$$
where $\vec{r_{m}}$ is the radius vector directed from the center of the circle *C_i_(x_ci_, y_ci_)* to the *m*-th pixel of the object with coordinates (*x_m_, y_m_*).

## 3. Results of the Analysis of the Number of Dimples of Tearing

In this paper, each image of the fracture surface was considered as a complexly organized system, which is characterized by a significant group and individual variations. This approach, in our opinion, allows us to adequately assess its geometric features, namely, to combine the common diagnostic features with stochasticity both in the structure of the stage and in the structure of the entire image, and to develop reliable methods for determining invariant informational features [28,29].

The input data for the analysis of the fractogram geometry were the discrete values of the diameters of dimples of ductile tearing within the area analyzed. They are fully determined by the zonal-spatial structure of the diameters of dimples of ductile tearing obtained by means of image segmentation methods. Let us consider the morphology of dimples typical of various magnifications:

х3000: Dimples measuring 2–6 μm were found. The predominant group was represented by dimples ≤2 μm. Such low dispersion of dimensions is inherent in this image, since dimples were analyzed on a very small area, not reached by large dimples.

х1200: Dimples measuring 2–10 μm were found. The predominant group was represented by dimples ≤4 μm. An increase in dispersion was due to an expansion of the analyzed area and the correct recognition of the edges of dimples.

х800: Dimples measuring 2–12 μm were found. The predominant group was represented by dimples ≤2 μm. Such high dispersion of dimensions is inherent in this image, since dimples were analyzed on a large area. The whole range of dimples inherent in the fracture surface of this specimen was present on it. In terms of analysis, this image was the most informative.

х200: Dimples measuring 2–6 μm were found. The predominant group was represented by dimples ≤2 μm. A decrease in dispersion is due to the fact that large dimples with edges decorated with smaller “micro-dimples” were not recognized.

The choice of an “optimal” increase in the surface analyzed had to provide for the detection of the entire range of dimples and allow their size to be estimated. The obtained distribution histograms of small dimples were almost non-sensitive to changes in the scale of the surface area analyzed.

It was found that the size distribution of dimples had an asymmetric view. This made it difficult to use mid-sized dimples to assess the most dangerous fracture mechanisms. The size of small dimples in the fracture is related to the size of grains and non-metallic inclusions, while large dimples, which had a more complex shape, resulted from the coalescence of pores. The results obtained were typical of high-strength materials characterized by a significant number of small dimples and only a few large dimples [30,31]. Such distribution of dimples evidences a combination of high strength and sufficient plasticity. However, the lack of large dimples in Figure 3d raises doubts as to the correctness of their recognition. It is clear that an increase in the dimple size reduces the likelihood of its falling into the fractogram with a large magnification, therefore, large dimples did not fall into the image, as shown in Figure 3a–c.

The algorithm used allows the distribution of intensity within the analyzed area to be taken into account and the geometry of individual dimples to be estimated, as shown in Figure 4a–c. In the context of fractographic control, using which the fracture of the analyzed VT23M alloy was considered as a set of two main components (dimples and their boundaries), the non-recognition of large dimples was found on fracture surfaces obtained at small magnifications, as shown in Figure 4d.

To evaluate the convergence of the results obtained for one plot at different magnifications, the recognized images of fractograms were overlapped, as shown in Figure 5. Figure 5a presents the superimposed results of detecting the edges of dimples with magnifications x1200 and x800, and Figure 5b presents the results obtained with magnifications x800 and x200. Approximately in the center of the image, a dimple is painted, which is used as a marker that allows comparison of the two images. A high degree of coincidence of the edges of dimples obtained with different magnifications was noted. A high degree of coincidence typical of the boundaries of recognized dimples was revealed, indicating the correctness of the size estimation of single dimples and their conglomerates, which is interpreted as isomorphism of the shape and size within a cyclic random process.

An in-depth analysis of fractograms and obtained results allowed us to schematize the analyzed dimples of tearing, as shown in Figure 6. We constructed a 2-D scheme of the plot of the analyzed fracture surface, which contains all types of dimples: A—individual dimples (small- and medium-sized); B—large dimples; C + D—large dimples, the surface of which is decorated with small dimples. These small dimples are located mainly on the walls of large dimples. It should be emphasized that the shape of these dimples is predominantly an irregular geometric one orientated towards the shift of the material. The geometry of these dimples is a source of information about the gradient of plastic deformations in the local macrodimple [32,33].

Given that the fracture surfaces are, in a way, “sewn” from a sequence of dimples of different sizes, which have typical segments, it was found that dimples of type A and B were recognized in full, dimples of type D were recognized completely, and dimples of type C were not recognized, because their boundaries were not recognized with magnification x200. As a result, these dimples were referred to as types A and B. Therefore, the approach used does not allow physically correct results to be obtained for dimples of type C + D, since it is focused on the recognition of individual objects. However, when used as the initial stage of the analysis, one can determine the smallest elements of the dimple structure and estimate its geometric dimensions.

The disadvantages of the proposed approach can be eliminated by using the brightness analysis of the local areas identified by the operator. This will provide for an additional in-depth analysis, which will make it possible to determine the contours of large dimples (C). All the small dimples that fell into these contours were considered as dimples of type D.

## 4. Identification of Large Dimples Decorated with Microdimples

An in-depth analysis consisted in detecting large dimples decorated with small dimples within the closed contour of a microdimple. At the physical level, it was necessary to identify a set of features and parameters that indicate the belonging of dimples to the C + D type.

To identify dimples, we used a previously developed and tested algorithm based on the convolutional neural network [19]. Then, we calculated the quantitative parameters of the resulting set of dimples [20]: The area, equivalent diameter, visual depth, slope, and number.

Surface fractograms obtained with different magnifications were studied. The peculiarity of the image obtained with a small magnification is that it contains large dimples, on the walls and the bottom of which there are many small dimples. If one focuses only on small dimples, the information about complex large dimple-like formations will be lost. Therefore, a slightly different approach was used for the analysis of such images (Figure 7). Initially, regions of large complex dimples were identified, and the distribution of small dimples within these large formations and beyond them were investigated separately.

To investigate the quantitative composition of large dimples (C + D), a crossover Gaussian filter with a large kernel size was first applied to the fractogram image $i_{0}$ (Figure 2a). As a result, an image was obtained that contained the topology of large dimples. Next, the thresholds were applied to it, resulting in a binary image mask $i_{m}$, which indicates places of the fractogram, on which large dimples are located. On the basis of this mask, image $i_{in}$ was formed, which represents the intersection of the initial fractogram with a mask and contains large dimples (Figure 8b):(3)$$i_{in}=i_{0}\cap^{\text{​}}i_{m}.$$

To estimate the parameters of small dimples, the calculations were repeated for the image outside the previously found large dimples. To do this, we highlighted image $i_{out}$ as an intersection of fractogram $i_{0}$ with an inverted mask ($\bar{i_{m}}$)
(4)$$i_{out}=i_{0}\cap^{\text{​}}\bar{i_{m}}.$$

Then, using the previously described algorithm [15], the quantitative parameters of the fractogram dimples located within the mask were investigated. According to the results of identifying dimples of type C + D, it is noticeable that the defects identified spatially correspond to the sections of the fractogram which can be classified by expert evaluation as large dimples decorated with microdimples. Within these areas, the number of micro-dimples was calculated, and their geometric parameters were determined.

Physical correctness of visualization of large dimples should be emphasized separately, because it allowed the quantitative characteristics of the dimples to be analyzed in detail. A good agreement between individual dimple parameters was revealed. However, the smallest increases (in our case) allow large surface areas to be covered and the geometry of large dimples to be taken into account, as shown in Figure 9a,b.

The curves shown in Figure 9b represent the dependence of the *K_c_* dimple parameter on its equivalent diameter. The resulting set of data for each image was approximated linearly by the least squares method. The angle of inclination of the obtained straight line to the vertical was determined. It should be emphasized that this parameter has a physical meaning: An increase in the parameter value indicates an increase in the integral approximation of the dimple shape of the analyzed set to the rounded one.

It was found that curves are located at an angle of 74° to the vertical, that is, the dimples are sufficiently rounded (Figure 9b). Further propagation and coalescence of dimples indicates their tendency to acquire a more elongated shape with an increase in size: for dimples larger than 10 µm, the shape coefficient acquires the fixed value of 55–60, indicating the effect of shear processes.

The rounded dimples, in our case, up to 6 µm, were formed in areas of structural heterogeneity, in particular, in the vicinity of inclusions, which was accompanied by the accumulation of local strains and fracture of the material. The consolidation of dimples and the distortion of their shape is primarily due to the transition to a volumetric stressed state, which causes the growth and coalescence of defects with the formation of large dimples [25,34]. It was believed that a change in the roundness coefficient *K_c_* indicates the localization of the dimple deformation at the microlevel due to the propagation of a crack formed as a result of coalescence of dimples (Figure 9a). Coefficient *K_c_* indicates the number of pixels entering the circle, the center of which is located in the center of mass of a circular concentrator with an equivalent diameter (Figure 9).

Since the cause of the formation of large dimples was not large inclusions, their presence in a fracture combined with thin connections between the inclusions indicates a significant influence of plastic deformations in the coalescence of adjacent pores after ductile fracture [14,25]. In general, after the application of the proposed approach, arrays of dimples >30 μm were found. These dimples were outside microdimples (microdimples are located in the larger 30 µm dimples) formed by the mechanisms of shear and separation, the shape of which is far from rounded [26,27,34].

These dimples caused a large scatter of the *K_c_* parameter, the value of which varies from 20 to 50.

The practical value of the data obtained is that the use of the mask to define the boundaries of large dimples has allowed to deepen the analysis of the fracture morphology of the titanium alloy VT23M, to reduce the influence of the scale factor for the analysis of the fractographic image, to ensure the physical and mechanical correctness of the results obtained.

## 5. Conclusions

This article is part of a series of publications on the study of fractographic images at various scale levels (macro, meso, micro) using the concept of physical mesomechanics. An approach is proposed and tested for the identification of informative features of dimples of ductile tearing, which in the case of individual dimples is invariant to the scale of measurement. At the core of the method for recognizing dimples is the use of the convolutional neural network. The offered approach allows the quantitative characteristics of objects of recognition to be obtained, which makes it possible to allocate quantitative classification signs and provide for their physical–mechanical and fractographic interpretation. Using fractograms with different magnifications allowed the morphological features of different types of dimples to be understood and described and their classification to be introduced, in particular: *A*—individual dimples (small and medium); *B*—large dimples; *C + D*—large dimples, the surface of which is decorated with small dimples.

The algorithm of analysis of fractographic images is proposed, taking into account the classification features of dimples of ductile tearing. Quantitative analysis of the dimple parameters was performed based on the calculation of their geometric parameters. The method for analyzing individual dimples *A, B* and large dimples *C* decorated with small dimples *D* is developed.

The proposed approaches are relevant for fractodiagnostics using a new generation of scanning microscopes with high resolution and the ability to scan extended areas of fracture surfaces. Even partial approbation of the proposed methods of analysis revealed new mechanisms of the localized plastic flow that affect the morphological characteristics of dimples of ductile tearing.

## Figures and Tables

**Figure 1 materials-12-02051-f001:**
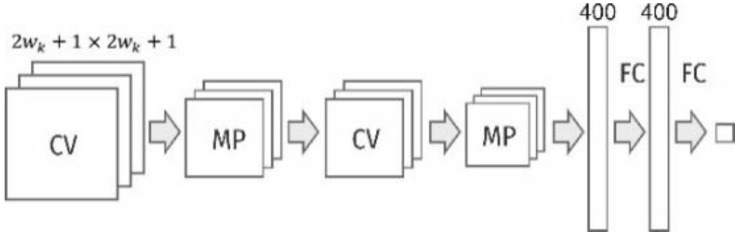
Architecture of the neural network used for the investigation of fractograms.

**Figure 2 materials-12-02051-f002:**
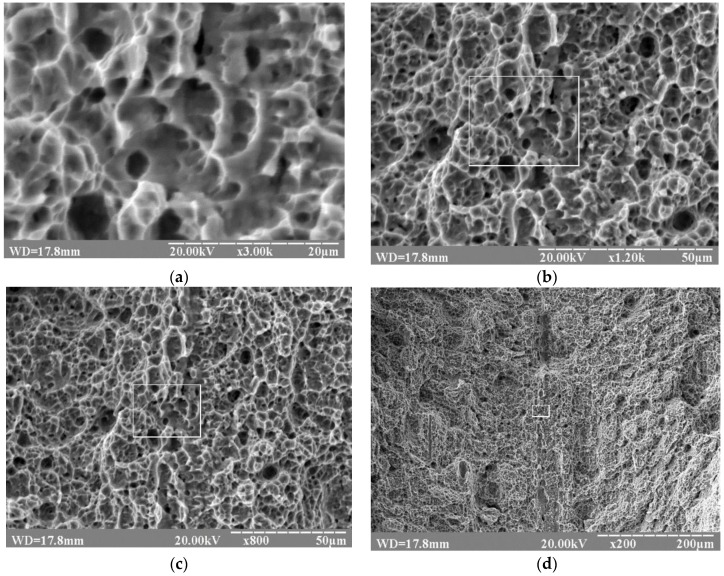
SEM images of fracture surfaces of specimens from the VT23M alloy at various magnifications: (**a**)—x3000; (**b**)—x1200; (**c**)—x800; (**d**)—x200; a frame on the fracture surface provides a link to the location of the fragment of the fracture surface shown in Figure 2a.

**Figure 3 materials-12-02051-f003:**
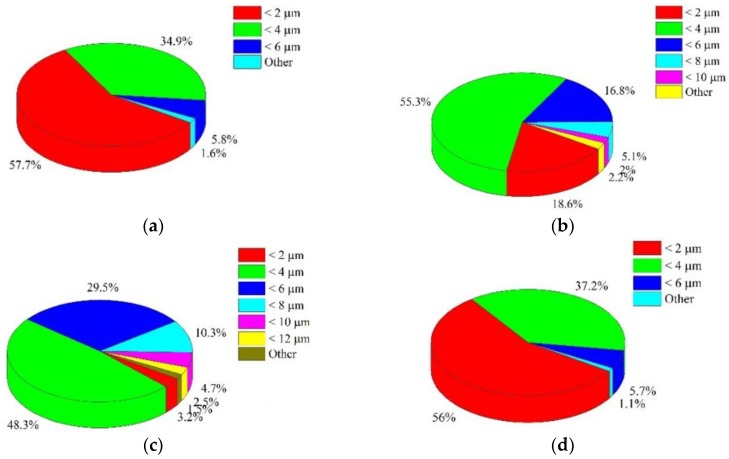
Specific distribution of dimensions of dimples found on the fracture surface of specimens from the VT23M alloy with the following magnifications: (**a**) x3000; (**b**) x1200; (**c**) x800; (**d**) x200.

**Figure 4 materials-12-02051-f004:**
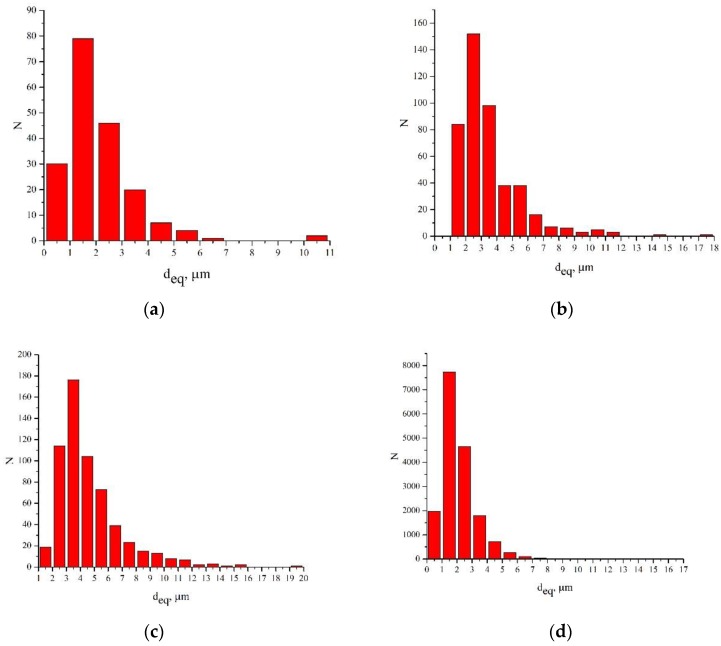
Histograms of equivalent diameters of dimples found on the fracture surface of specimens from the VT23M alloy with the following magnifications: (**a**) x3000; (**b**) x1200; (**c**) x800; (**d**) x200.

**Figure 5 materials-12-02051-f005:**
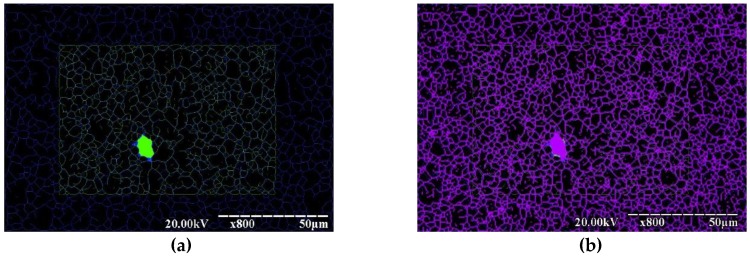
Graphical comparison of the network of dimples identified in fractograms with different magnification. Overlapping of dimple networks identified with magnifications x1200 and x800 (**a**); overlapping of dimple networks identified with magnifications x800 and x200 (**b**).

**Figure 6 materials-12-02051-f006:**
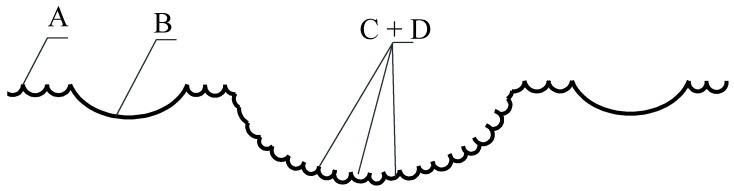
2-D scheme of the identified dimples and their classification: A—individual dimples (small- and medium-sized); B—large dimples; С + D—large dimples (С), the surface of which is decorated with small dimples (D).

**Figure 7 materials-12-02051-f007:**
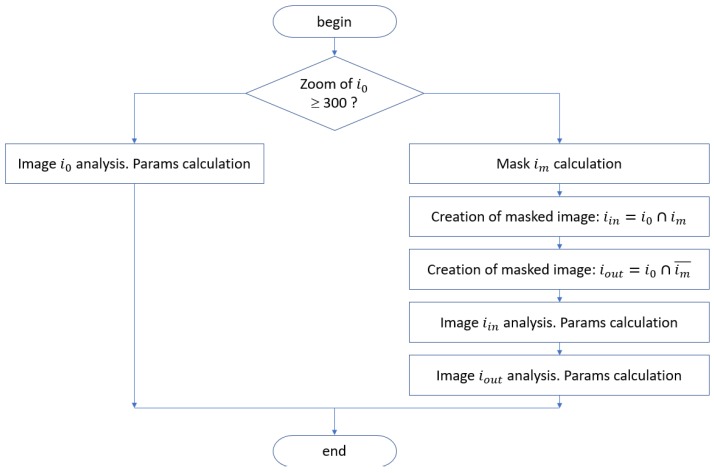
Algorithm for analysis of fractograms of titanium alloy VT23M depending on their magnification.

**Figure 8 materials-12-02051-f008:**
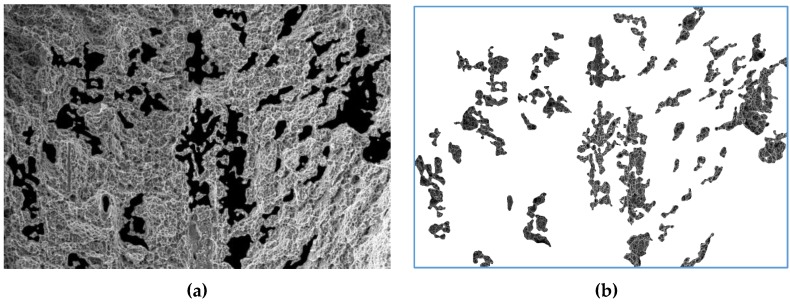
Fractogram of the fracture surface (Figure 2d) of specimens from alloy VT23M: Fractogram sections highlighted by a mask (**a**); image i_in with large dimples, inside of which there are small dimples (**b**); fragment of the obtained image with dimples recognized inside large dimples (**c**); image i_out with the surface outside large dimples (**d**).

**Figure 9 materials-12-02051-f009:**
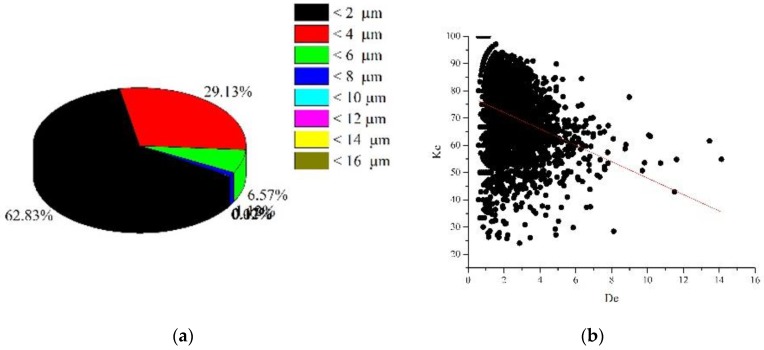
Distribution of sizes—(**a**); $K_{c}$ dimple shape parameter—(**b**); and $d_{e}$ equivalent diameter—(**c**) for the fracture surface (Figure 2d) of alloy VT23M.

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
