# Peer review of "Morphological Characteristics of Dimples of Ductile Fracture of VT23M Titanium Alloy and Identification of Dimples on Fractograms of Different Scale"

_materials, 2019, doi:10.3390/ma12132051_

Round 1
Reviewer 1 Report
This work develops a method for the automated detection and calculation of quantitative parameters of ductile fracture dimples on the digital images of a titanium alloy fracture surfaces. The research is interesting, however, to be accepted for publication the following comments are required to be addressed:
1- The introduction needs to be improved on titanium alloys, their properties and applications. Please read and use the following papers: Materials Science and Engineering: A 593 (2014) 170-177, Materials & Design 76 (2015), 47-54, Materials & Design 111 (2016), 592-599.
2- Images in Fig. 1 are required to be reordered from lower magnification to higher magnification (i.e. Fig. 1(d) needs to be Fig. 1(a), …).
3- Line 99 needs to be modified to “equivalent diameter (di), visual depth, inclination,”.
4- In the experimental section, in Line 99, please specify the parameters which are going to be calculated instead of putting “so on”.
5- Please put scale bar in Fig. 4 (a,b).
Author Response
ANSWER TO REVIEWER 1
1- The introduction needs to be improved on titanium alloys, their properties and applications. Please read and use the following papers: Materials Science and Engineering: A 593 (2014) 170-177, Materials & Design 76 (2015), 47-54, Materials & Design 111 (2016), 592-599
Titanium alloys are used for the protective outfit of soldiers (plates of body armor are made from them), as well as implants, surgical instruments, internal and external prosthetics, including such critical high-tech elements as heart valves. In addition, titanium alloys are used for the manufacture of aircrafts and space vehicles.
Attar H., Calin M., Zhang L.C., Scudino S., Eckert J. (2014). Manufacture by selective laser melting and mechanical behavior of commercially pure titanium. Materials Science and Engineering A, 593, 170-177, doi: 10.1016/j.msea.2013.11.038
Haghighi S.E., Lu H.B., Jian G.Y., Cao G.H., Habibi D., Zhang L.C. Effect of α″ martensite on the microstructure and mechanical properties of beta-type Ti-Fe-Ta alloys, Materials and Design, (2015) 76 , pp. 47-54, doi:10.1016/j.matdes.2015.03.028
Ehtemam-Haghighi, S., Prashanth, K. G., Attar, H., Chaubey, A. K., Cao, G. H. and Zhang, L. C. (2016). Evaluation of mechanical and wear properties of Ti-xNb-7Fe alloys designed for biomedical applications. Materials and Design 111 592-599. https://doi.org/10.1016/j.matdes.2016.09.029
2- Images in Fig. 1 are required to be reordered from lower magnification to higher magnification (i.e. Fig. 1(d) needs to be Fig. 1(a), …).
Fractographic studies were conducted based on the concept of mesomechanics, that is, from the micro to macro level [11]. Changing the order of the fractogram analysis may destroy the structure of scientific interpretation of the results, so we cannot follow this recommendation.
3- Line 99 needs to be modified to “equivalent diameter (di), visual depth, inclination,”.
Corrected.
4- In the experimental section, in Line 99, please specify the parameters which are going to be calculated instead of putting “so on”.
Corrected.
5- Please put scale bar in Fig. 4 (a,b).
Corrected.
Reviewer 2 Report
The method is proposed for the automated detection and calculation of quantitative parameters of dimples of ductile fracture of fracture surfaces. At the core of the method for recognizing dimples is the use of convolutional neural network. However, the processing algorithm is not presented in the paper, and the training set and the test set are also not very clear. There is no result comparisons between the proposed method and other different methods, and the accuracy is not discussed as well.
Author Response
ANSWER TO REVIEWER 2
The method is proposed for the automated detection and calculation of quantitative parameters of dimples of ductile fracture of fracture surfaces. At the core of the method for recognizing dimples is the use of convolutional neural network. However, the processing algorithm is not presented in the paper, and the training set and the test set are also not very clear. There is no result comparisons between the proposed method and other different methods, and the accuracy is not discussed as well. The method is proposed for the automated detection and calculation of quantitative parameters of dimples of ductile fracture of fracture surfaces. At the core of the method for recognizing dimples is the use of convolutional neural network. However, the processing algorithm is not presented in the paper, and the training set and the test set are also not very clear. There is no result comparisons between the proposed method and other different methods, and the accuracy is not discussed as well.
Our current article is a continuation of the study, which is highlighted in a number of previous papers [18-20].
Konovalenko, I.; Maruschak, P.; Chausov, M.; Prentkovskis, O. Fuzzy logic analysis of parameters of dimples of ductile tearing on the digital image of fracture surface. Proc. Engin. 2017, 187, 229-234, doi:10.1016/j.proeng.2017.04.369
Konovalenko, I.; Maruschak, P.; Prentkovskis, O.; Junevičius, R. Investigation of the rupture surface of the titanium alloy using convolutional neural networks. Materials 2018, 11, 2467, ; https://doi.org/10.3390/ma11122467
Maruschak, P.; Konovalenko, I.; Chausov, M.; Pylypenko, A.; Panin, S.; Vlasov, I.; Prentkovskis, O. Impact of dynamic non-equilibrium processes on fracture mechanisms of high-strength titanium alloy VT23. Metals 2018, 8, 983, https://doi.org/10.3390/met8120983.
Therefore, the authors did not begin describing the features of recognizing fractographic images from the first stage, because these features are already known. The highlighted results should be considered using references to previous articles contained in the text of this article. To aid in the understanding of the algorithm features, the text of the article was supplemented.
The structure of the neural network, the functions of neuron activation, the optimizer, hyperparameters, as well as the learning process of the proposed neural network are described in detail in [19]: https://www.mdpi.com/1996-1944/11/12/2467/htm
For training of the neural network where used 562,856 image fragments measuring 31 × 31 pixels. Test dataset consist of 134,596 ones. In previous work 17 models of neural networks with various sets of hyperparameters have been developed and investigated. Than their accuracy and speed were investigated, and the optimal neural network model was selected. The proposed network contained two convolution layers, two subsampling layers, and two fully connected layers. The neural network classified the pixels into two categories: “dimples” and “edges”, and admitted the affiliation of each pixel to one of these classes. Also we have compared results of dimples recognition by proposed network with results obtained by known algorithm for analyzing fractograms. It has been found that the results are very similar (more than 90% similarity), but the neural network reveals the necessary features more accurately than the previous method.
Reviewer 3 Report
I appreciate the effort to get a quantitative description of the ductile fracture surface, an din particular to quantify th size distribution of dimples to be correlated to the mechanical properties of the material.According to my opinion, the authors should improve the description of the results, by making clear reference to the figures in the taxt. For instance, the results shown in fig. 2 and 3 are described (lines 120 to 136) before to introduce and show the figures. Moreover, figure 4 should be described with spme more detail.
Line 255-256: I cannot inderstand how arrays of dimples bigger than 30 micrometers can be inside microdimples.
Three point remarks.
authors use approximation in place of magnification: why ?
line 123: authors mention morphology (let us consider morphology), but they describe size of dimples. Please edit.
lines 249-251: the meaning of Kc is already explained at page 4. Please remove.
Author Response
ANSWER TO REVIEWER 3
For instance, the results shown in fig. 2 and 3 are described (lines 120 to 136) before to introduce and show the figures. Moreover, figure 4 should be described with spme more detail
An explanation has been added to the article.
Line 255-256: I cannot inderstand how arrays of dimples bigger than 30 micrometers can be inside microdimples Line 255-256: I cannot inderstand how arrays of dimples bigger than 30 micrometers can be inside microdimples
This is a well-known feature, which is well described in the literature on fractography [26, 27]. The authors used the well-known physical-mechanical concept regarding the formation of dimples of tearing. In case of the coalescence of pores, the shape of which is not ideal, the ductile tearing occurs unevenly. Therefore, in the place of microdisperse phases, the surfaces of large dimples are covered with micro-dimples.
authors use approximation in place of magnification: why ?
Corrected.
line 123: authors mention morphology (let us consider morphology), but they describe size of dimples. Please edit.
In this article, dimples are the main element of the surface of the failure, therefore, in our opinion, we can assert that we investigate the morphology of the surface of the fracture, because we consider its main components. From a physico-mechanical point of view, our statement is correct.
lines 249-251: the meaning of Kc is already explained at page 4. Please remove. lines 249-251: the meaning of Kc is already explained at page 4. Please remove.
Corrected.
Round 2
Reviewer 1 Report
The authors have addressed the comments and the quality of the manuscript has been improved. Therefore, the work can be accepted for publication.
Author Response
Thank You kindly for Your review.
Reviewer 2 Report
The reviewer did not find any significant improvements from the old version, please find the comments and suggestions in the previous review report.
Author Response
Answer to the issue has been presented in previous reply. Also corrected version of manuscript has been submitted.